# Spatio-Temporal Niche Differentiation of Alpine Musk Deer, Chinese Serow, and Tufted Deer in Changdu Prefecture, Tibet, China

**DOI:** 10.3390/biology14111536

**Published:** 2025-11-03

**Authors:** Changjian Wang, Yang Yu, Yang Liu, Tong Zhang, Fu Shu, Yuling Chen, Jiyuan Yu, Yi Chen, Haochun Chen, Zhuoma Quzhen, Ali Krzton, Keji Guo, Zuofu Xiang

**Affiliations:** 1Institute of Evolutionary Ecology and Conservation Biology, Central South University of Forestry & Technology, Changsha 410004, China; wangchangjian1949@163.com (C.W.); yuyangjcl@126.com (Y.Y.); 17704465187@163.com (Y.L.); ychen1009@163.com (Y.C.); 2College of Forestry, Central South University of Forestry & Technology, Changsha 410004, China; 3Hunan Agricultural Forestry and Industrial Prospective Design Institute Co., Ltd., Changsha 410004, China; 4College of Life Science and Technology, Central South University of Forestry & Technology, Changsha 410004, China; 5Central South Academy of Inventory and Planning of NFGA, Changsha 410014, China; zhangtong0429@163.com (T.Z.); shufu328qs@163.com (F.S.); chenhc_zny@126.com (H.C.); guokeji@126.com (K.G.); 6PowerChina Zhongnan Engineering Corporation Limited, Changsha 410004, China; 03586@msdi.cn; 7Huaneng Lancang River Hydropower Inc., Kunming 650214, China; ayuan755@163.com; 8Mangkang Forestry and Grassland Bureau, Mangkang 854000, China; mkxlcj@126.com; 9Department of Innovation and Research Support, RBD Library, Auburn University, Auburn, AL 36849, USA; iamthekat@gmail.com

**Keywords:** *Elaphodus cephalophus*, *Moschus chrysogaster*, *Capricornis milneedwardsii*, Changdu, spatio-temporal niche

## Abstract

**Simple Summary:**

Understanding how species coexist is essential for conserving biodiversity. This study investigates the spatio-temporal niche differentiation among three sympatric ungulates—alpine musk deer (*Moschus chrysogaster*), Chinese serow (*Capricornis milneedwardsii*), and tufted deer (*Elaphodus cephalophus*)—in Changdu, Tibet. Using data from 186 camera traps and species distribution models (SDMs), we analyzed their habitat preferences and daily activity rhythms. The results revealed distinct patterns: Tufted deer occupied smaller, fragmented forest habitats (total suitable area: 14,968 km^2^), while alpine musk deer (41,909 km^2^) and Chinese serow (36,954 km^2^) showed broader distributions with significant spatial overlap (26,869 km^2^). Temporally, tufted deer and alpine musk deer exhibited diurnal activity with peak overlaps (Δ = 0.88), whereas chinese serow was primarily nocturnal, reducing direct competition through temporal partitioning (Δ = 0.76–0.82; all *p* < 0.001). These findings underscore how spatial and temporal niche differentiation facilitates coexistence, providing a scientific basis for targeted conservation strategies in high-altitude ecosystems.

**Abstract:**

Knowledge of the mechanisms of species coexistence is crucial for biodiversity conservation. The tufted deer (*Elaphodus cephalophus*), alpine musk deer (*Moschus chrysogaster*), and Chinese serow (*Capricornis milneedwardsii*) are all found in alpine ecosystems in Tibet, China. To investigate how these sympatric species achieve stable coexistence, we compared species distribution models and diurnal activity rhythms to analyze their spatial and temporal niche characteristics based on data collected from 186 camera traps set in Changdu prefecture. The results indicate he following: (1) In Changdu, the total area of suitable habitats for tufted deer, alpine musk deer and Chinese serow are 14,968 km^2^, 41,909 km^2^, and 36,954 km^2^, respectively. These areas represent approximately 13.62%, 38.14%, and 33.63% of the study area, respectively. (2) The total overlapping area of suitable habitats between tufted deer and alpine musk deer is 5102 km^2^. The total overlapping area of suitable habitats between tufted deer and Chinese serow is 6483 km^2^. Additionally, the total overlapping area of suitable habitats between alpine musk deer and Chinese serow amounts to 26,869 km^2^. (3) The overlap index for daily activity rhythms between tufted deer and alpine musk deer is 0.88—this difference is statistically significant (*p* < 0.001). Similarly, the overlap index for daily activity rhythms between tufted deer and Chinese serow is 0.82—also significantly different (*p* < 0.001). Lastly, the overlap index for daily activity rhythms between alpine musk deer and Chinese serow is 0.76—again showing a significant difference (*p* < 0.001). The results provide valuable insight into conservation strategies aimed at preserving populations and habitats of tufted deer, alpine musk deer, and Chinese serow while contributing to a deeper understanding of resource partitioning mechanisms as well as population dynamics among coexisting species.

## 1. Introduction

Species coexistence is one of the central issues in ecological research. Understanding the mechanisms of stable coexistence is crucial for conserving biodiversity and maintaining ecosystem functions. The niche of a species is primarily composed of resources, enemies, time, and space [1]. Niche differentiation plays a key role in allowing species to coexist over time [2]. Animals reduce interference or competition by altering their windows of activity within the day or month [3]. Investigating species that share the same habitat offers a perspective on how organisms coexist, thereby advancing the theoretical framework of community ecology. Such studies also provide foundational knowledge for conservation planners, enabling them to formulate more effective management approaches for nature reserves and protected zones [4].

Tufted deer (*Elaphodus cephalophus* Milne-Edwards, 1872), alpine musk deer (*Moschus chrysogaster* Hodgson, 1839), and Chinese serow (*Capricornis milneedwardsii* David, 1869) belong to Cetartiodactyla: Cervidae, Moschidae, and Bovidae, respectively. It is noteworthy that all of these species have recognized subspecies [5]. The tufted deer and Chinese serow are classified as Class II nationally protected species in China, while the alpine musk deer is a Class I nationally protected animal. The tufted deer’s conservation status is “near threatened” on the IUCN Red List and “vulnerable” according to China’s national assessment [6,7]. An ecologically significant species, the alpine musk deer has been designated a protected Class I animal in China and “endangered” on the IUCN Red List, functioning as a solitary ruminant crucial to mountainous ecosystems [8,9]. Finally, classified as “vulnerable” by the IUCN, serows (*Capricornis* spp.) constitute essential herbivores inhabiting subtropical and tropical forest ecosystems across Southeast Asia [10,11]. These three ungulates collectively contribute to biodiversity preservation and ecological balance through their distinct roles in forest vegetation dynamics and nutrient cycling processes.

In recent studies on species coexistence and niche analysis, species distribution models (SDMs) have become a core tool for quantifying habitat suitability and spatial niche partitioning, with the Maximum Entropy (MaxEnt) model being the most widely used framework—this is rooted in its strong performance in handling presence-only data and integrating complex environmental variables, as demonstrated in Phillips et al.’s foundational study on MaxEnt for species geographic distribution modeling [12]. For instance, Alanís-Méndez et al. applied MaxEnt combined with WorldClim climate data to assess the impact of climate change on the distribution of the endemic Mexican orchid *Prosthechea mariae* and the effectiveness of protected areas (PAs), revealing that PAs may lose 36–48% of suitable habitat for the orchid by 2025 [13]. This not only confirms MaxEnt’s reliability in predicting habitat shifts but also provides a reference for our study to link spatial niche results with conservation practice. Similarly, Liu et al. optimized MaxEnt parameters (e.g., feature types and regularization multipliers) to evaluate the suitable habitat of Tufted deer in Hupingshan National Nature Reserve, finding that the wettest quarter precipitation and coldest quarter mean temperature contribute 56% to habitat selection—this directly informs our selection of key environmental variables for ungulate SDMs, given the shared focus on deer species [14].

For model refinement and multi-scenario validation, Moo-Llanes used the kuenm package in R to calibrate MaxEnt models for predicting the invasive potential of the Asian giant hornet *(Vespa mandarinia*) in the Americas, emphasizing that screening optimal parameter combinations (e.g., regularization multipliers of 0.1–10) reduces overfitting [15]. Beyond terrestrial species, Rodriguez-Burgos et al. used kuenm-optimized MaxEnt to predict the distribution shift of the common hammerhead shark (*Sphyrna lewini*) in the tropical eastern Pacific under RCP2.6 and RCP8.5 scenarios, proving SDMs’ universality across terrestrial and marine ecosystems—this further validates our use of MaxEnt for terrestrial ungulate niche analysis [16]. For invasive or sympatric species, Werenkraut et al. integrated Global Biodiversity Information Facility (GBIF) data and literature occurrences with MaxEnt to identify suitable invasion areas of the Oriental hornet (*Vespa orientalis*) in the Americas, highlighting that multi-source occurrence data reduce bias in niche modeling [17]. Yang et al. further systematically reviewed MaxEnt’s application in species habitat prediction, noting its advantage in integrating multiple environmental variables to capture niche characteristics—this supports our inclusion of both climatic (e.g., annual mean temperature) and topographic (e.g., elevation) variables in SDMs [18].

Beyond modern distribution modeling, historical climate change has long been recognized as a key driver of species niche evolution and coexistence patterns. Defalque et al. investigated the impact of drought and high temperatures on plant-pollinator interactions in common buckwheat (*Fagopyrum esculentum*), finding that combined stress reduces nectar volume by 45–70% and pollen production by 33–70%, indirectly altering the foraging niche of herbivores dependent on these plants [19]. This provides a reference for interpreting how historical resource fluctuations (e.g., Pleistocene drought events) shaped the foraging niche differentiation of our study’s three ungulates, which rely on forest vegetation for food. Lipiec et al. reviewed the combined effects of drought and heat stress on plant growth and yield, noting that long-term climate fluctuations (e.g., glacial-interglacial cycles) alter the temporal and spatial availability of herbivore resources (e.g., plant phenology shifts), further driving interspecific niche differentiation [20]. This aligns with our focus on historical climate’s role in shaping current coexistence patterns.

Hamann et al. revealed that climate change modifies plant-herbivore interactions by altering plant defense traits (e.g., increased secondary metabolites under heat stress) and herbivore feeding preferences [21]. For our study, this implies that historical climate shifts may have driven the three ungulates to specialize in different plant species or parts (e.g., alpine musk deer favoring understory herbs vs. serows favoring woody browse), promoting niche partitioning. Felton et al. [22] systematically analyzed the response of deer species in boreal and temperate regions to climate change, finding that warmer winters benefit population growth but hotter summers exceed physiological tolerances.

Paleontological and phylogeographic data further complement the understanding of long-term species coexistence mechanisms. Litvinchuk et al. used MaxEnt niche modeling combined with molecular markers to identify glacial refugia and post-glacial colonization routes of three cryptic marsh frog species (*Pelophylax* spp.) in the Western Palearctic [23].

To investigate how these three sympatric deer species maintain stable coexistence in Changdu, we integrated species distribution models (SDMs) with daily activity rhythm analyses using camera-trap data from 186 locations. This multidimensional niche analysis revealed distinct spatio-temporal resource partitioning patterns among the species, providing empirical evidence for their coexistence mechanisms. Our findings not only advance understanding of population dynamics in sympatric ungulates but also offer critical insights for habitat conservation and modeling resource partitioning mechanisms in mountainous ecosystems. Specifically, understanding the coexistence mechanisms of these vulnerable species is essential for developing targeted conservation strategies, predicting their responses to environmental changes, and effectively managing the fragile high-altitude ecosystems they inhabit.

## 2. Materials and Methods

### 2.1. Study Area

Changdu prefecture (93°6′–99°2′, 28°5′–32°6′), located in the eastern part of the Tibet Autonomous Region, covers an area of about 19,870 km^2^. It borders the provinces of Qinghai, Sichuan, and Yunnan to the north, east, and southeast, respectively. In Changdu, the great fold systems of the Himalayas and the northern Nyainqêntanglha Mountains swing southeast, forming a series of high parallel ranges with a predominantly northwest-to-southeast axis. Between these rows of mountains, the upper streams of the Nu, Lancang, and Jinsha rivers flow from northwest to southeast through deep, forested chasms. The topography of the mountain system is characterized by steep elevations and towering peaks, with the highest point exceeding 5000 m (Figure 1). Most of the area is uninhabited, and large parts remain virtually unexplored. There are many tributaries that provide sufficient water for wildlife. The climate of Changdu is subtemperate, with an annual temperature of about 7.6 °C and an average annual precipitation of about 400–600 mm. Suitable geographical and climatic conditions have created rich biodiversity. There are many wild animals distributed in the mountains, including tufted deer, alpine musk deer, Chinese serow, black-and-white snub-nosed monkey (*Rhinopithecus bieti*), leopard (*Panthera pardus*), snow leopard (*Panthera uncia*), brown bear (*Ursus arctos*), blood pheasant (*Ithaginis cruentus*), and white eared pheasant (*Crossoptilon crossoptilon*) [24].

### 2.2. Spatial Niche Analysis

#### 2.2.1. Occurrence Points of Species

Between August 2023 and August 2024, we set 40 camera-trap sites using infrared-triggered cameras (models #E3 Series Camera, EREAGLE, Shenzhen, China; model #BG962-X36W, BolyMedia, Shenzhen, China) in the Karuo zone, Changdu prefecture, Tibet, China. Between July 2021 and July 2023, we set 99 camera-trap sites using infrared-triggered cameras (models #BG962-X36W, BolyMedia, Shenzhen, China; model #SG560KV, Shenzhen, China) in Luolong county, Changdu prefecture, Tibet, China. Between December 2021 and December 2022, we set 47 camera-trap sites using infrared-triggered cameras (models #BG962-X36W, BolyMedia, Shenzhen, China; #E3 Series Camera, EREAGLE, Shenzhen, China; model #Ere-E1, EREAGLE, Shenzhen, China; model #SG-999V, Bestguarder, Shenzhen, China) in Mangkang county, Changdu prefecture, Tibet, China. We installed all cameras on trees along forest roads approximately 1 m above the ground. We programmed the cameras to capture three images for each trigger, with no delay between each trigger. We identified the focal species from the three images and recorded the species, camera-trap site, data, and time as a single detection point. We removed the recurring points within a rectangular area of 1 km^2^ to mitigate spatial autocorrelation during the model operation process [25]. The spatially distributed data filter “Spatially Rarefy Occurrence Data for SDMs” (reduce spatial autocorrelation) in the SDM toolbox was used to dilute the data within 1 km of other data points [26]. After filtering, 25 tufted deer, 72 alpine musk deer, and 76 Chinese serow distribution points remained (see Figure 2).

#### 2.2.2. Environmental Variables and Data Processing

Three types of environmental variables (vegetation, terrain, and climate) were selected for the model construction (Table 1). The vegetation dataset consists of three components: normalized difference vegetation index (NDVI), land cover classification, and forest tree height. Specifically, NDVI and land cover data were obtained from the National Qinghai–Tibet Plateau Scientific Data Center (https://data.tpdc.ac.cn) (accessed on 25 October 2025), while forest tree height measurements were sourced from the 3D Ecology Platform (https://www.3decology.org (accessed on 25 October 2025)). Nineteen bioclimatic factors were selected from the World Climate Database (https://www.worldclim.org). Topographic factors, including elevation, slope, and aspect, are derived from the Digital Terrain Elevation Model (DEM) and obtained through the geospatial data cloud platform of the Computer Network Information Center at the Chinese Academy of Sciences (https://www.gscloud.cn (accessed on 25 October 2025)). The same geographic extent, grid cell size (1 km^2^), and projected coordinate system (WGS 1984 UTM Zone 47N) were selected for all layers. As it is difficult to interpret the model output, especially the relative importance of the variables and their response curves, when the environmental variables used to train the model are highly correlated [27], we used the R 4.3.3 “SDMtune” package for constructing MaxEnt models. In SDMtune, variable selection was performed using a built-in function that pairs the correlated variables using Jackknife tests, retaining only the variable with the highest contribution for each pair. The “SDMtune” package was chosen for its efficient built-in function “varSel”, which systematically reduces multicollinearity among predictor variables using Jackknife tests, thereby enhancing model interpretability and performance [28]. Finally, seventeen environmental variables were screened into the MaxEnt models.

#### 2.2.3. Mapping the Suitable Habitat of Species

The MaxEnt model’s configuration primarily depends on two parameters: the regularization multiplier (RM) and feature classes (FCs), which operate in conjunction with species occurrence data and corresponding environmental predictors [29]. A suitable combination of these elements can raise the sensitivity of the model while lowering the risk of overfitting. The parameters of the MaxEnt model were adjusted and optimized through the “ENMeval” package (version 2.0.4) in R 4.3.3, aiming to identify the most suitable model configurations for each species studied. The “ENMeval” package was employed because it provides a robust framework for automatically tuning key MaxEnt parameters (regularization multiplier and feature classes), which helps in identifying optimal model settings that balance model fit and complexity, thus reducing overfitting [30,31]. The modeling framework incorporated georeferenced species presence data and bioclimatic predictors, partitioned through random allocation into a 75% subset for parameter calibration and a 25% subset for predictive validation. To enhance model reliability, 10 iterative bootstrap simulations were conducted across randomized data partitions, with final predictions derived from ensemble averaging of all computational iterations [32]. According to the maximum training sensitivity plus specificity (MTSS) and balance training omission, predicted area and threshold value (BTPT), the habitats were divided into non-suitable habitat, sub-suitable habitat, and highly suitable habitat. We chose MTSS and BTPT for thresholding because: (1) MTSS balances sensitivity (ability to correctly identify suitable habitats) and specificity (ability to exclude non-suitable areas), which is critical for protecting rare species like the study’s focal ungulates; (2) BTPT minimizes training omission while controlling the predicted suitable area, avoiding overestimation of habitat extent—a common problem when using single thresholds (e.g., 10% training presence) for high-altitude species with fragmented habitats. Quantitative measures of predictive accuracy are essential to assessing model performance. Among available validation metrics, the area under the receiver operator curve (AUC) has become the predominant measure in species distribution modeling studies using MaxEnt approaches [33].

#### 2.2.4. Niche Overlap Analysis

Methodological selection, contingent upon both data structural characteristics and targeted research questions, plays a vital role in ensuring precise quantification of niche overlap [34]. In the present study, ENMTools 1.4.4 was used to calculate Schoener’s D (D) and Hellinger’s-based I (I) values to compare the niche overlap of tufted deer, alpine musk deer, and Chinese serow [35]. ENMTools was selected for two key reasons: first, it directly integrates with MaxEnt output to calculate niche overlap metrics, avoiding errors associated with manual data conversion; second, Schoener’s D and Hellinger’s I are widely used metrics in ungulate niche overlap studies, ensuring comparability of the present results with those of existing literature.

### 2.3. Temporal Niche Analysis

#### 2.3.1. Data Processing

Wild animal species captured in the infrared camera photographs from Changdu were identified and classified. We utilized “camtrapR” (version 2.2.0) in R to efficiently manage and analyze our camera trap data [36]. The “camtrapR” package is designed for camera trap data processing, particularly for projects handling large volumes of data. It supports the complete workflow of camera trap data management, including image organization, annotation, species and individual identification, image metadata extraction, tabulation and visualization of results, and data export for further analysis [37]. The photographs of wildlife were systematically classified by species and timestamped with precise capture data. Each image was further categorized as either dependent or independent based on specific criteria. Independent events encompassed three distinct scenarios: consecutive captures showing different individuals regardless of species identity, sequential images of the same species separated by over 30 min, and non-consecutive photographic records of conspecific animals. The 30 min interval was selected to define independent events, a standard criterion in camera-trap studies that effectively minimizes temporal autocorrelation while accounting for potential animal residency time near the camera [38]. In total, 4345 infrared camera photos of tufted deer were utilized, with 1375 images being independently and effectively recorded. Additionally, 3106 photos of alpine musk deer were employed, among which 1372 were recorded independently and effectively. Finally, 1291 photos of Chinese serow were analyzed, with 443 images being independently and effectively captured.

#### 2.3.2. Analysis of Daily Activity Rhythm

Random samples were drawn from the set of independent and effective detections of tufted deer, alpine musk deer, and Chinese serow. A daily activity rhythm model was developed utilizing kernel density estimation to analyze the characteristics and interactions of the daily activity patterns among tufted deer, alpine musk deer, and Chinese serow. This approach posits that the likelihood of capturing target species via infrared cameras during a given timeframe correlates directly with their movement frequency. Each valid photographic record constitutes an independent sampling event that follows the statistical characteristics of the species’ detection probability distribution. Consequently, the derived probability density function of photographic occurrences serves as an effective indicator for characterizing wildlife activity patterns across daily light-dark cycles, reflecting temporal variations in animal behavior [39]. A comparison of the nuclear density curves among tufted deer, alpine musk deer, and Chinese serow was conducted in parallel to estimate the degree of symmetric overlap in their daily activity rhythms. This overlap ranged from 0, indicating no overlap, to 1, indicating complete overlap. Analysis of animal daily activity patterns from infrared camera data employed kernel density estimation utilizing the “overlap” (version 0.3.3) and “activity” (version 1.3.3) packages in R. The “overlap” package was specifically designed for calculating kernel density estimates and overlap coefficients from circular time data, making it ideal for analyzing temporal activity patterns [40]. The “activity” package was used for its robust statistical tests to compare activity patterns between species, as it accounts for the circular nature of time data [41]. The temporal activity patterns of tufted deer, alpine musk deer, and Chinese serow were analyzed using R 4.3.3. Kernel density estimation and overlap coefficient calculations were performed with the “overlap” package to quantify temporal niche sharing among species. Comparative analysis of daily activity rhythms was conducted using the “activity” package, with statistical significance determined at *p* < 0.05. To assess daily activity rhythms between species, Δ1 was employed for those with fewer than 75 records, while Δ4 was computed for pairs with at least 75 records, with all analyses implemented through the “overlap” R package [40,41]. All data processing, statistical evaluations, and graphical visualizations utilized the R programming environment.

### 2.4. Consideration of Sample Size Limitations

While our sample sizes for alpine musk deer (*n* = 72) and Chinese serow (*n* = 76) are robust for species distribution modeling, the sample size for tufted deer (*n* = 25) is relatively small. We acknowledge that smaller sample sizes can increase model uncertainty and potentially affect the predictive accuracy and transferability of the model. To mitigate this limitation, we employed the following strategies: (1) Using a relatively large number of bootstrap replicates (10 iterations) to stabilize model predictions; (2) Utilizing a simple model configuration (via ENMeval) to avoid overfitting; and (3) Interpreting the results for tufted deer with appropriate caution, focusing on general patterns rather than fine-scale predictions. The models for all species showed acceptable performance based on AUC values, yet the findings for tufted deer should be considered as preliminary and would benefit from additional data in future studies.

## 3. Results

### 3.1. Spatial Niche Analysis

#### 3.1.1. Prediction of Suitable Habitat

The MaxEnt model result showed that the AUC values of tufted deer, alpine musk deer, and Chinese serow were 0.994, 0.950, and 0.957, respectively, which indicated that the predictions of the MaxEnt model were accurate and can be used for habitat analysis. The maximum training sensitivity plus specificity (MTSS) of the MaxEnt model results were 0.1176, 0.2709, and 0.2402, respectively, and the mean value of the balance training omission, predicted area, and threshold value (BTPT) was 0.0129, 0.0726, and 0.0705, respectively. According to the above threshold, the MaxEnt model prediction results were divided into highly suitable habitat, sub-suitable habitat, and unsuitable habitat. Detailed metrics of model performance and suitable habitat area/proportion for each species are presented in Table 2.

The suitable habitats for the tufted deer were mainly distributed in southern Changdu, with a small portion in the north (Figure 3). However, the suitable habitats for the alpine musk deer and Chinese serow were primarily located in Changdu prefecture, with a small proportion in the south.

#### 3.1.2. Impact of Environmental Factors on Habitat Quality

Habitat suitability models for tufted deer, alpine musk deer, and Chinese serow are strongly influenced by key environmental variables. For tufted deer, the five most significant factors included mean temperature of the warmest quarter (Bio18), annual temperature range (Bio7), land cover, normalized difference vegetation index (NDVI), and forest tree height, collectively contributing 98% to the model. Alpine musk deer habitat suitability was primarily shaped by temperature seasonality (Bio4), forest tree height, mean temperature of the warmest quarter (Bio18), elevation, and annual temperature range (Bio7), with a cumulative contribution of 74%. Similarly, Chinese serow distribution was most affected by precipitation during the wettest quarter (Bio16), forest tree height, mean temperature of the wettest quarter (Bio8), isothermality (Bio3), and temperature seasonality (Bio4), determining 79.9% of the model.

Tufted deer favor habitats lacking grasslands, where warmest-quarter precipitation (Bio18) is below 320 mm and annual temperature variation (Bio7) remains under 30 °C, showing a marked preference for forested areas with high NDVI and taller trees (see Figure 4). This suggests a strong reliance on dense forest cover for thermoregulation, shelter from predators, and access to understory food resources, while avoiding open grasslands and areas with high temperature fluctuations or excessive summer rainfall. In contrast, alpine musk deer thrive in regions with temperature seasonality (Bio4) exceeding 660, warmest-quarter precipitation (Bio18) under 360 mm, and annual temperature range (Bio7) below 32 °C, typically below 4600 m elevation with taller forest canopies (see Figure 5). These conditions likely represent areas with sufficient climatic variability that support diverse vegetation, while the preference for taller forests provides crucial cover from predators and harsh weather at high altitudes. Chinese serow habitats are associated with wettest-quarter precipitation (Bio16) under 350 mm, mean wettest-quarter temperatures (Bio8) above 8 °C, isothermality (Bio3) below 44%, and temperature seasonality (Bio4) between 650 and 750, reflecting a reliance on specific climatic stability and seasonal rainfall patterns (see Figure 6). This implies a preference for thermally stable habitats with moderate monsoon rainfall, which may favor the growth of their preferred forage and reduce physiological stress compared to regions with extreme temperature diurnal ranges or excessive wet-season precipitation.

#### 3.1.3. Habitat Overlaps

The spatial niche overlap among the three species exhibited distinct patterns, as detailed in Table 3. In brief, the overlap between tufted deer and the other two species was relatively low, whereas a very high degree of spatial niche overlap was observed between alpine musk deer and Chinese serow. The high spatial overlap between alpine musk deer and Chinese serow, coupled with significant temporal niche partitioning (see Section 3.2), suggests that these sympatric species may reduce direct competition through differential activity timing. Conversely, the lower spatial overlap of the tufted deer with the other species indicates stronger habitat segregation, which likely minimizes competitive interactions.

### 3.2. Temporal Niche Analysis

Both tufted deer and alpine musk deer exhibited high levels of activity during daylight hours. In contrast, the Chinese serow demonstrated primarily nocturnal activity patterns. The two activity peaks of tufted deer were more obvious around 9:00 and 19:00. The two activity peaks of alpine musk deer were more obvious around 9:00 and 20:00. The two activity peaks of Chinese serow were more obvious around 5:00 and 21:00. In terms of temporal niche overlap, the overlap index for the daily activity rhythms of tufted deer and alpine musk deer was 0.88 (Figure 7), with the Wald test indicating the difference between these two species was statistically significant (*p* < 0.001). The overlap index of the daily activity rhythm of tufted deer and Chinese serow was 0.82, with Wald test results showing a significant difference between them (*p* < 0.001). The overlap index of the daily activity rhythm of alpine musk deer and Chinese serow was 0.76, and Wald test results showed that there was a significant difference between them (*p* < 0.001).

## 4. Discussions

When competing species coexist, less effective competitors often employ temporal niche partitioning by shifting their activity patterns to minimize overlap with dominant species [42]. This behavioral adaptation may be complemented by spatial avoidance strategies, where inferior competitors preferentially utilize suboptimal habitats to reduce encounters with dominant species in preferred resource areas [43]. The present study revealed distinct patterns of ecological niche differentiation among the three focal ungulate species across multiple dimensions. Notably, the tufted deer diverged from both the alpine musk deer and Chinese serow in both temporal and spatial niche dimensions [44,45]. In contrast, the alpine musk deer and Chinese serow exhibited extensive spatial overlap. However, they displayed significant temporal segregation in their activity patterns.

Habitat differentiation, recognized as the most prevalent and ecologically significant manifestation of niche partitioning among sympatric species [46], can occur easily where there is pronounced spatial stratification along altitudinal gradients. Extensive empirical evidence confirms that altitudinal environmental variation serves as a critical filter shaping vertical stratification patterns in wildlife assemblages. Species distributions exhibit elevational niche specialization mediated by thermal tolerances, resource availability, and interspecific competition [47,48]. In this study, alpine musk deer were predominantly observed at elevations below 4000 m. Studies have shown that habitat selection in wildlife often involves trade-offs between concealment and other ecological factors [49]. This pattern is exemplified by the distribution preferences of tufted deer and alpine musk deer, whose presence shows a positive correlation with increasing tree height. This relationship likely reflects the enhanced concealment benefits provided by taller vegetation, which offers improved protection from predators and reduced human disturbance in forest ecosystems. Research has indicated that body size variations also support the ecological coexistence of populations of sympatric species [50]. In this context, the Chinese serow, being the largest of the three focal species, likely occupies a distinct ecological niche. Fossil and isotopic evidence further support this ecological niche continuity: Suraprasit et al. reported Late Middle Pleistocene fossils of Capricornis spp. from Southeast Asia, which were associated with montane forest habitats nearly identical to the forest-dominated, high-altitude environment of the current study [51]. Suraprasit et al. further verified this adaptation using stable carbon isotope (δ^13^C) analysis of *Capricornis sumatraensis* (a congener of *C. milneedwardsii*) fossils, showing that this genus relied primarily on C_3_ plants—typical of closed forest ecosystems—and confirmed a long-term association with humid, vegetation-dense habitats [52]. Its larger size may enable access to tougher vegetation, greater foraging ranges, or alternative microhabitats, thereby minimizing direct competition with smaller sympatric ungulates and supporting ecological partitioning in shared environments.

When the spatial niches of sympatric species significantly overlap, these species tend to differentiate along alternative niche axes such as temporal patterns or resource utilization strategies. This multidimensional niche segregation minimizes interspecific competition while facilitating ecological coexistence [53]. Temporal niche partitioning may manifest through diurnal/nocturnal activity differentiation or seasonal variations in reproductive and foraging behaviors, further optimizing resource allocation among co-occurring species. Activity patterns and temporal niche partitioning demonstrate how organisms manage temporal constraints. Research on wildlife diurnal activity cycles provides insight into ecological behavioral adaptations [54] and interspecies interactions [55]. These biological rhythms are modulated by multiple factors, including nutritional requirements, predation pressure, reproductive cycles, anthropogenic disturbances, meteorological variations, and thermal conditions [56]. The alpine musk deer and tufted deer demonstrate convergent patterns in their circadian activity rhythms. Both species exhibit predominantly diurnal behavior with distinct crepuscular tendencies, characterized by bimodal activity peaks during dawn and dusk. Their daily cycle typically follows a specific pattern: first, a surge in activity during early morning hours; then, reduced movement intensity during midday thermal maxima; finally, a second pronounced active period before twilight. The observed bimodal activity pattern in alpine species, characterized by peaks during crepuscular periods, likely serves as an adaptive strategy to mitigate the detrimental effects of harsh solar radiation exposure while minimizing energetic expenditure [57]. In contrast, the Chinese serow exhibited strictly nocturnal activity patterns. This montane ungulate maintains peak mobility throughout the night, with behavioral observations indicating optimal foraging and locomotion occurring under the cover of darkness. Such temporal niche differentiation likely represents evolutionary adaptations that minimize interspecific competition and facilitate predator avoidance in shared habitats. Empirical evidence indicates that predation risk promotes strategic temporal adjustments in both predators and prey, resulting in dynamic avoidance mechanisms [58].

The results provide empirical data of use for protecting tufted deer, alpine musk deer, and Chinese serow while serving as a framework for high-altitude wildlife conservation [59]. Its findings have significant implications for preserving biodiversity and maintaining ecosystem stability in Changdu, Tibet. Specifically, understanding multi-dimensional niche differentiation is crucial for predicting species responses to environmental changes, such as climate change and human encroachment. This knowledge allows managers to develop targeted conservation strategies. For instance, if climate change alters vegetation zones along the altitudinal gradient, understanding the specific habitat requirements and spatial partitioning among these species can help in designing habitat corridors or prioritizing areas for protection. Similarly, if human activities shift the activity patterns of one species, managers can anticipate potential increased temporal overlap and competition with other sympatric species like the nocturnal Chinese serow, and take mitigating measures. Therefore, our results underscore the importance of a multi-dimensional approach to wildlife management, which considers the complex interplay between niche axes to safeguard sympatric species assemblages under changing environmental conditions. However, methodological constraints and data limitations necessitate cautious interpretation of results. The predictive accuracy of species distribution models is inherently influenced by data quantity, spatial distribution characteristics, and study scale parameters [60]. Insufficient sample sizes, uneven spatial sampling, and inappropriate geographic delineation may introduce estimation biases, potentially causing localized overestimation or underestimation of habitat suitability [61,62].

In the application of species distribution models (SDMs), methodological and data constraints remain key challenges to result interpretation: statistical biases in functional response modeling for habitat selection, such as the impact of random intercepts on coefficient adjustment and sensitivity to poorly estimated coefficients under scarce resources, may lead to misjudgments of habitat preference dynamics [63]. Meanwhile, SDM performance varies with spatial scale and data sources—fine-scale local data improves current distribution prediction accuracy but tends to cause niche truncation, which affects the reliability of future projections [64]. For endemic or near-threatened species, SDMs often identify key driving factors (e.g., annual precipitation, elevation) for suitable habitats, yet unsampled peripheral fragmented regions easily introduce localized estimation biases, and future climate scenarios may further induce habitat contraction [65]. Additionally, anthropogenic disturbances (e.g., agricultural expansion, infrastructure development) synergize with climate change to exacerbate habitat fragmentation; human activities alone can significantly reduce suitable habitat areas, and together with climate-driven range shifts, they amplify deviations in model estimations [66].

## 5. Conclusions

Our study quantified the habitat characteristics of three sympatric ungulates. The total area of suitable habitat in Changdu was 14,968 km^2^ for tufted deer, 41,909 km^2^ for alpine musk deer, and 36,954 km^2^ for Chinese serow. The models identified the top five contributing environmental factors for each species, which were primarily related to temperature, precipitation, and forest structure (e.g., forest tree height). The niche overlap was highest between alpine musk deer and Chinese serow (Hellinger’s I = 0.97, Schoener’s D = 0.83; overlapping suitable habitat area = 26,869 km^2^), and lower between tufted deer and the other two species. Conversely, the daily activity rhythms showed significant overlap for all species pairs (Δ > 0.76, *p* < 0.001). These findings reveal the patterns of niche partitioning and coexistence among the three species. More importantly, they provide a scientific basis for developing targeted conservation strategies. For instance, the high habitat similarity between alpine musk deer and Chinese serow allows for integrated conservation planning, while the distinct requirements of tufted deer necessitate focused habitat protection. We recommend that these insights be incorporated into regional habitat management and species protection plans to enhance conservation effectiveness.

## Figures and Tables

**Figure 1 biology-14-01536-f001:**
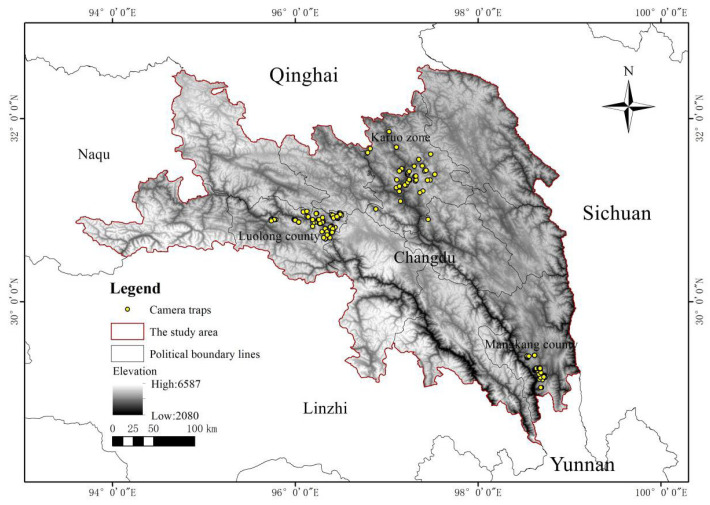
Maps of the study area showing camera trap sites relative to Changdu.

**Figure 2 biology-14-01536-f002:**
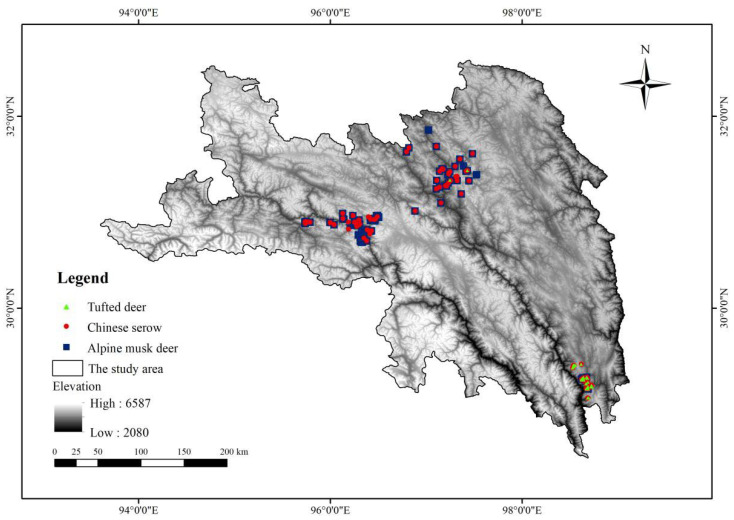
Species occurrence records for tufted deer, alpine musk deer, and Chinese serow.

**Figure 3 biology-14-01536-f003:**
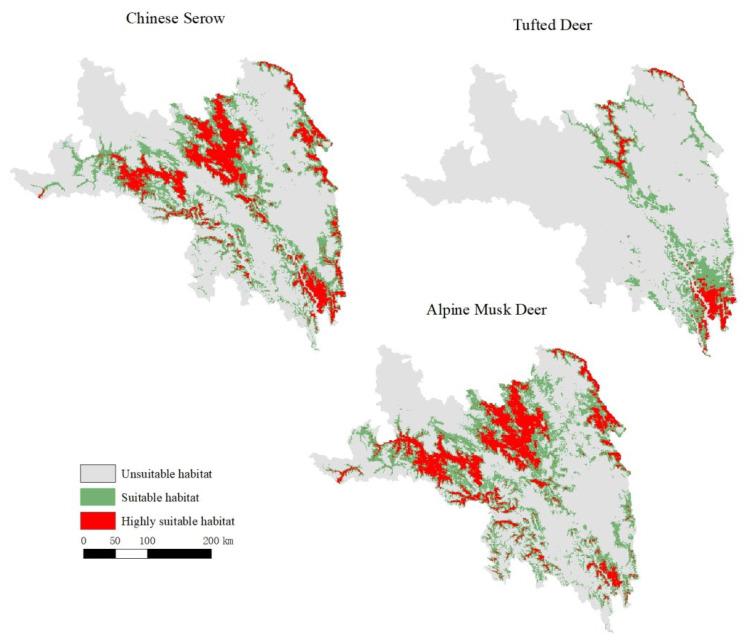
Suitable habitat of tufted deer, alpine musk deer, and Chinese serow.

**Figure 4 biology-14-01536-f004:**
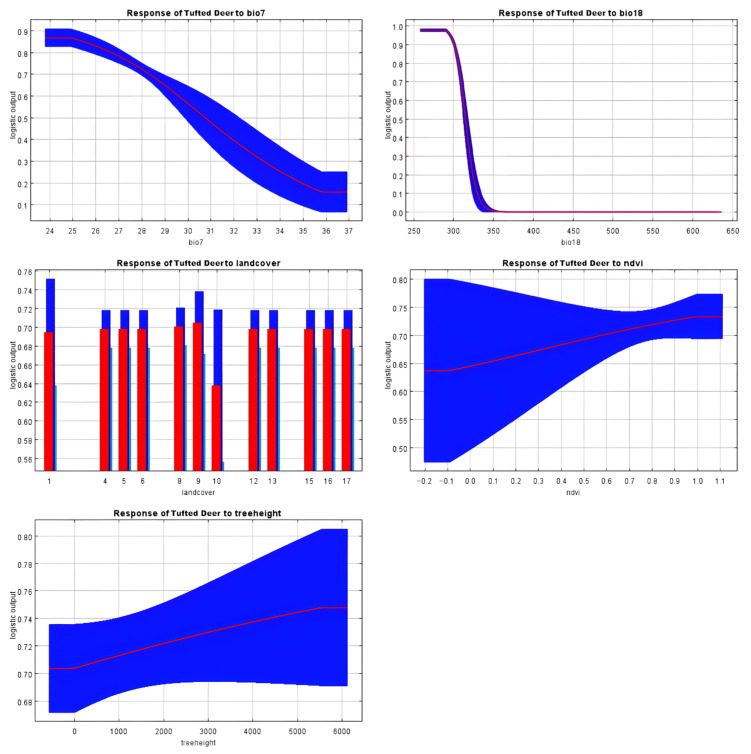
Response curve results of MaxEnt modeling of the first five environmental variables for the suitable habitat for tufted deer. Note: Land cover products in the subplot titled “Response of Tufted Deer to landcover” were classified into 17 categories defined by the International Geosphere Biosphere Programme (IGBP). The specific category codes and corresponding descriptions are as follows: 1 = Evergreen Needleleaf Forests; 2 = Evergreen Broadleaf Forests; 3 = Deciduous Needleleaf Forests; 4 = Deciduous Broadleaf Forests; 5 = Mixed Forests; 6 = Closed Shrublands; 7 = Open Shrublands; 8 = Woody Savannas; 9 = Savannas; 10 = Grasslands; 11 = Permanent Wetlands; 12 = Croplands; 13 = Urban and Built-Up Lands; 14 = Cropland/Natural Vegetation Mosaics; 15 = Permanent Snow and Ice; 16 = Barren; 17 = Water Bodies.

**Figure 5 biology-14-01536-f005:**
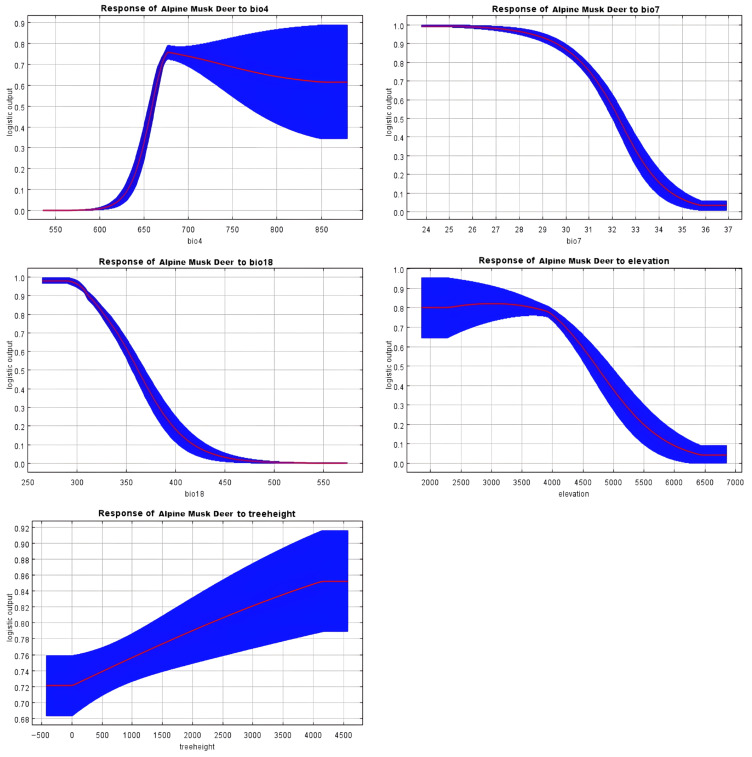
Response curve results of MaxEnt modeling of the first five environmental variables for the suitable habitat for alpine musk deer.

**Figure 6 biology-14-01536-f006:**
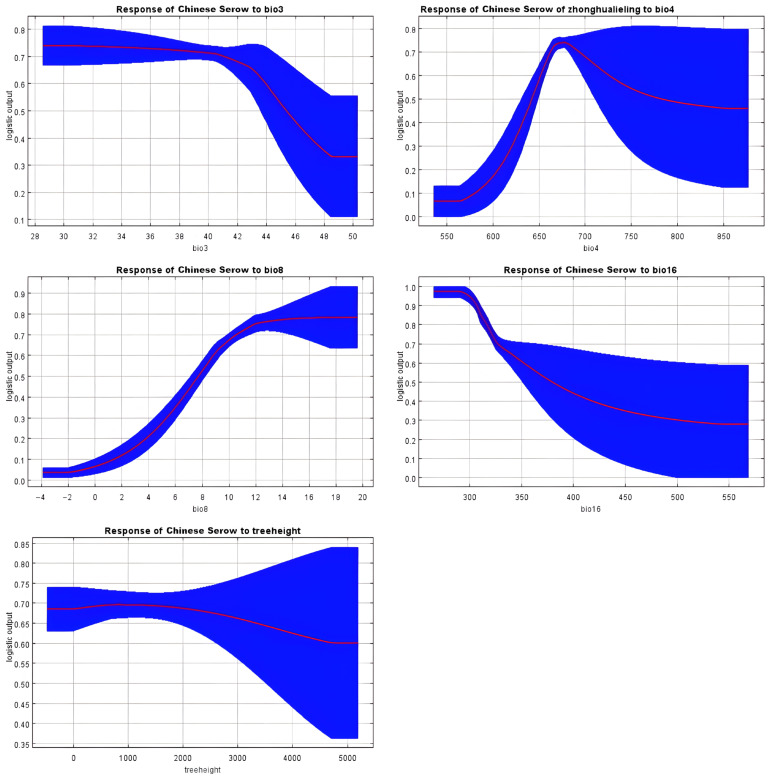
Response curve results of MaxEnt modeling of the first five environmental variables for the suitable habitat for Chinese serow.

**Figure 7 biology-14-01536-f007:**
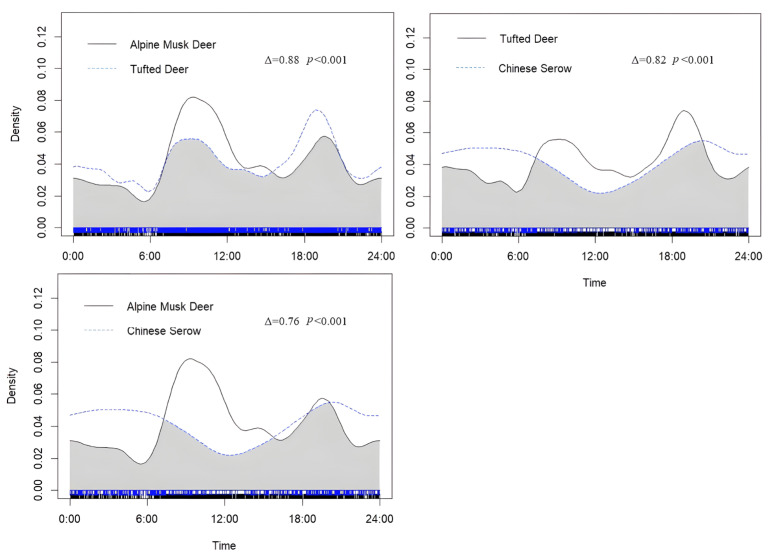
The daily activity rhythms of tufted deer, alpine musk deer, and Chinese serow.

**Table 1 biology-14-01536-t001:** Environmental variables used in the analysis for tufted deer (T), alpine musk deer (A), and Chinese serow (C); abbreviations: T = Tufted deer, A = Alpine musk deer, C = Chinese serow.

Environmental Variables	Source	Type of Variable	Applied Species
Bio2	http://www.worldclim.org (accessed on 25 October 2025)	Continuous variables	C
Bio3	http://www.worldclim.org (accessed on 25 October 2025)	Continuous variables	T, A, C
Bio4	http://www.worldclim.org (accessed on 25 October 2025)	Continuous variables	A, C
Bio7	http://www.worldclim.org (accessed on 25 October 2025)	Continuous variables	T, A, C
Bio8	http://www.worldclim.org (accessed on 25 October 2025)	Continuous variables	C
Bio10	http://www.worldclim.org (accessed on 25 October 2025)	Continuous variables	T
Bio12	http://www.worldclim.org (accessed on 25 October 2025)	Continuous variables	T, C
Bio14	http://www.worldclim.org (accessed on 25 October 2025)	Continuous variables	T, A, C
Bio15	http://www.worldclim.org (accessed on 25 October 2025)	Continuous variables	A, C
Bio16	http://www.worldclim.org (accessed on 25 October 2025)	Continuous variables	C
Bio18	http://www.worldclim.org (accessed on 25 October 2025)	Continuous variables	T, A
Aspect	http://www.gscloud.cn (accessed on 25 October 2025)	Continuous variables	T, A, C
Elevation	http://www.gscloud.cn (accessed on 25 October 2025)	Continuous variables	A
Land cover	https://data.tpdc.ac.cn (accessed on 25 October 2025)	Discrete variables	T, A, C
NDVI	https://data.tpdc.ac.cn (accessed on 25 October 2025)	Continuous variables	T, A, C
Slope	http://www.gscloud.cn (accessed on 25 October 2025)	Continuous variables	T, A, C
Forest tree height	https://www.3decology.org (accessed on 25 October 2025)	Continuous variables	T, A, C

**Table 2 biology-14-01536-t002:** MaxEnt Model Evaluation and Suitable Habitat Statistics for Three Ungulate Species.

Species	Model Evaluation Indices	Habitat Type	HSI Threshold	Area, km^2^	Proportion of Study Area, %
Tufted Deer	AUC = 0.994	Highly Suitable	>0.1176	3147	2.86
	MTSS = 0.1176	Sub-suitable	0.0129–0.1176	11,821	10.76
	BTPT = 0.0129	Total Suitable	>0.0129	14,968	13.62
Alpine Musk Deer	AUC = 0.950	Highly Suitable	>0.2709	15,016	13.67
	MTSS = 0.2709	Sub-suitable	0.0726–0.2709	26,893	24.48
	BTPT = 0.0726	Total Suitable	>0.0726	41,909	38.14
Chinese Serow	AUC = 0.957	Highly Suitable	>0.2402	13,860	12.61
	MTSS = 0.2402	Sub-suitable	0.0705–0.2402	23,094	21.02
	BTPT = 0.0705	Total Suitable	>0.0705	36,954	33.63

**Table 3 biology-14-01536-t003:** Habitat overlap metrics and niche overlap indices among three ungulate species.

Species Pair	Total Overlap Area, km^2^	Highly Suitable Overlap Area, km^2^	Proportion of Own Total Habitat, %		Proportion of Own Suitable Habitat, %		Niche Overlap Indices	
			Species A	Species B	Species A	Species B	*D*	*I*
Tufted deer–Alpine musk deer	5102	1658	34.09	12.17	11.08	3.96	0.3098	0.5983
Tufted deer–Chinese serow	6483	2455	43.31	17.54	16.4	6.64	0.3889	0.6999
Alpine musk deer–Chinese serow	26,869	11,242	64.11	72.71	26.82	26.82	0.8266	0.9729

Notes: In each pair, “Species A” refers to the first-listed species, and “Species B” refers to the second-listed species; “Proportion of own total habitat” is the percentage of overlapping area relative to the total habitat area of the species; “Proportion of own suitable habitat” is the percentage of highly suitable overlapping area relative to the suitable habitat area of the species.

## Data Availability

The datasets used in this study are available from the corresponding author upon reasonable request.

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
