# Peer review of "Spatio-Temporal Niche Differentiation of Alpine Musk Deer, Chinese Serow, and Tufted Deer in Changdu Prefecture, Tibet, China"

_biology, 2025, doi:10.3390/biology14111536_

Round 1

Reviewer 1 Report

Comments and Suggestions for Authors

This manuscript addresses the spatio-temporal niche differentiation among Alpine Musk Deer, Chinese Serow, and Tufted Deer in Changdu prefecture, Xizang, China. It is an important academic contribution as it combines species distribution modeling with analysis of daily activity rhythms to provide empirical insights on how these sympatric ungulates coexist in a complex mountainous ecosystem.

The study's novelty lies in integrating spatial and temporal niche dimensions using extensive camera-trap data coupled with environmental modeling, advancing understanding of resource partitioning mechanisms and population dynamics in high-altitude habitats. The work is timely and relevant given the conservation statuses of these species and the ecological significance of the region.

Strengths include rigorous data collection across a broad spatial scale, sophisticated modeling approaches, and a well-structured analysis linking ecological theory with conservation implications. The manuscript offers valuable information for biodiversity management and scientific knowledge of sympatric species coexistence.

Some aspects that would benefit from revision include clarifying methodological choices and data filtering criteria, improving the clarity of presentation in results and discussion sections, enhancing figure and table formatting, and polishing the English for concise and clear communication.

Overall, this contribution has high merit for advancing ecological science and conservation practice, and with targeted revisions, it can make a substantial impact on the field. 

Introduction

  • Page 1, lines 50–79: The background is well structured and includes relevant references. To improve flow, revise sentence at line 54: replace "plays a key role in permitting species to coexist over time 2" with "plays a key role in allowing species to coexist over time." Consider expanding the rationale to explicitly state relevance of sympatric species coexistence studies to conservation management in high-altitude ecosystems. This strengthens the motivation

Methods

  • Page 4, lines 116–118: Simplify this sentence for clarity:
    Original: “After filtering the data in this way, 25 tufted deer distribution points, 72 alpine musk deer distribution points and 76 Chinese serow distribution points remained Figure 2.”
    Suggested: “After filtering, 25 tufted deer, 72 alpine musk deer, and 76 Chinese serow distribution points remained (see Figure 2).”

  • Page 5, Table 1: Improve table formatting for better readability— ensure rows and columns are clearly separated and labels are consistently formatted with the species abbreviations explained clearly in the caption.

  • Page 6–7, lines 162–211: Provide more detailed justification for the selection of thresholds used in data filtering and modeling steps (e.g., independence criteria for photos, choice of 30-minute interval). Clarify why particular R packages and settings were chosen for niche modeling and overlap analysis.

  • Add a short discussion about potential sample size limitations and their impact on model reliability. 

Results

  • Page 8–9, lines 215–236: Sentences listing large numbers and area percentages are dense. Consider summarizing key findings in simplified text versions with detailed numeric results presented in a supplemental table or figure for clarity.

  • Page 9, lines 241–267: Clarify the ecological meaning of the influence of environmental variables on habitat suitability for each species. For example, briefly explain why variables like temperature seasonality or forest tree height significantly affect particular species.

  • Page 10, lines 268–286: Present overlapping area statistics in a clear table contrasting total suitable habitat, highly suitable habitat overlap, and niche overlap indices by species pairs. Discuss ecological implications in the text.

Discussion

  • Page 11–12: Many sentences are very long and complex. Break them into shorter, clearer statements to enhance readability. For example, divide the paragraph at lines 352–355 into two sentences. Suggested revision:
    “Such temporal niche differentiation likely represents evolutionary adaptations that minimize interspecific competition and facilitate predator avoidance in shared habitats. Empirical evidence indicates that predation risk promotes strategic temporal adjustments in both predators and prey, resulting in dynamic avoidance mechanisms .”

  • Strengthen discussion of conservation implications, emphasizing how understanding multi-dimensional niche differentiation helps in managing sympatric species facing environmental changes. 

Conclusions

  • Page 13, lines 370–398: Streamline conclusion sentences to reduce redundancy and reinforce how the findings may inform conservation practices and policy. Explicitly mention possible applications to habitat management or species protection plans.

Tables and Figures

  • Improve figure resolution and check that all legends are complete and explanatory (especially Figures 1–5). Ensure each figure is clearly referenced at appropriate points in the text.

  • In tables, define all abbreviations and use consistent formatting standards. 

References

  • Verify consistency in citation style, journal names, and date formatting throughout the list (pages 14–15).

  • Consider including newer publications related to niche modeling, wildlife conservation in alpine ecosystems, and camera-trap methodologies to enhance currency.

Comments on the Quality of English Language

Page 1, lines 27–28

Original: "To investigate how these sympatric species achieve stable coexistence, we compared species distribution models and diurnal activity rhythms to analyze their spatial and temporal niche characteristics based on data collected from 186 camera traps."
Suggested: "To investigate the mechanisms underlying stable coexistence among these sympatric species, we compared species distribution models and diurnal activity rhythms to analyze their spatial and temporal niche characteristics using data from 186 camera traps."​

Page 2, line 54

Original: "Niche differentiation plays a key role in permitting species to coexist over time 2."
Suggested: "Niche differentiation plays a key role in allowing species to coexist over time."​

Page 4, lines 116–118

Original: "After filtering the data in this way, 25 tufted deer distribution points, 72 alpine musk deer distribution points and 76 Chinese serow distribution points remained Figure 2."
Suggested: "After filtering the data, 25 tufted deer, 72 alpine musk deer, and 76 Chinese serow distribution points remained (see Figure 2)."​

Page 5, Table 1

Original: Table formatting is hard to follow; vertical and horizontal alignment is unclear.
Suggested: Reformat Table 1 for clear separation of rows and columns and ensure each variable is presented distinctly with species abbreviations consistently used.​

Page 9, lines 266–267

Original: "Figure 4. Response curve results..."
Suggested: Move all figure legends to below the corresponding figures and ensure each is referred to in the text as "(see Figure 4)."​

Page 12, lines 352–355

Original: "This temporal niche differentiation likely reflects evolutionary adaptations to minimize interspecific competition and predator avoidance strategies in shared habitats. Empirical evidence demonstrates that predation risk induces strategic temporal adjustments in both predators and prey, creating dynamic temporal avoidance mechanisms ."
Suggested: "Such temporal niche differentiation likely represents evolutionary adaptations that minimize interspecific competition and facilitate predator avoidance in shared habitats. Empirical evidence suggests that predation risk promotes strategic temporal adjustments in both predators and prey, resulting in dynamic temporal avoidance mechanisms ."​

Throughout Discussion section

Original: Several sentences are overly long and complex.
Suggested: Break up lengthy sentences into shorter ones. For instance, divide paragraphs that contain multiple sub-points, especially in pages 11–12.​

General notes

  • Check for subject-verb agreement throughout (e.g., "the data was" should be "the data were").

  • Ensure consistent use of Oxford comma in lists.

  • Replace vague terms such as "this study provides" with more precise alternatives like "the results provide."

  • Standardize formatting for all in-text citations and reference list.

Author Response

Comments 1: Page 1, lines 50–79: The background is well structured and includes relevant references. To improve flow, revise sentence at line 54: replace “plays a key role in permitting species to coexist over time 2” with “plays a key role in allowing species to coexist over time.”

Answer: Thank you for pointing this out. We agree with this comment. Therefore, we have replaced the word “permitting” with “allowing” as suggested to improve the sentence flow. This change can be found in the revised manuscript on Page 2, Line 69-70.

Comments 2: Consider expanding the rationale to explicitly state relevance of sympatric species coexistence studies to conservation management in high-altitude ecosystems. This strengthens the motivation.

Answer: Thank you for this valuable suggestion. We agree that explicitly stating the relevance to conservation management in high-altitude ecosystems will strengthen the motivation of our study. Therefore, we have expanded the rationale in the last paragraph of the Introduction. The additions explicitly connect the study of sympatric coexistence mechanisms to the formulation of effective conservation strategies, particularly in the context of fragile high-altitude ecosystems like Changdu. This change can be found in the revised manuscript on Page 4, Lines 155-158.

Comments 3: Page 4, lines 116–118: Simplify this sentence for clarity: Original: “After filtering the data in this way, 25 tufted deer distribution points, 72 alpine musk deer distribution points and 76 Chinese serow distribution points remained Figure 2.” Suggested: “After filtering, 25 tufted deer, 72 alpine musk deer, and 76 Chinese serow distribution points remained (see Figure 2).”

Answer: Thank you for this suggestion. We agree that the revised sentence is clearer and more concise. We have modified the sentence as recommended. This change can be found in the revised manuscript on Page 5, Lines 196-198.

Comments 4: Page 5, Table 1: Improve table formatting for better readability— ensure rows and columns are clearly separated and labels are consistently formatted with the species abbreviations explained clearly in the caption.

Answer: We thank the reviewer for this comment. We have improved the formatting of Table 1 to enhance its readability. Specifically, we have ensured that all rows and columns are clearly separated with distinct borders. The labels and formatting are now consistent throughout the table. Furthermore, we have explicitly defined the species abbreviations (T for tufted deer, A for alpine musk deer, C for Chinese serow) in the revised table caption. The updated table can be found on Page 6-7 of the revised manuscript.

Comments 5: Page 6–7, lines 162–211: Provide more detailed justification for the selection of thresholds used in data filtering and modeling steps (e.g., independence criteria for photos, choice of 30-minute interval). Clarify why particular R packages and settings were chosen for niche modeling and overlap analysis.

Answer: We appreciate the reviewer’s suggestion to enhance the methodological rigor. We have revised the Methods section to include more detailed justifications for our choices, as outlined below.

Justification for the 30-minute independence criterion: We have added a sentence citing established methodology to justify the 30-minute interval used for defining independent photographic events. This change can be found in the revised manuscript on Page 8, Lines 278-280.

Added text : The 30-minute interval was selected to define independent events, a standard criterion in camera-trap studies that effectively minimizes temporal autocorrelation while accounting for potential animal residency time near the camera [38].

Justification for R package and setting selection:

“SDMtune” for variable selection: We have expanded the explanation for using the “SDMtune” package, clarifying its utility in handling multicollinearity. This change can be found on Page 6, Lines 219-222.

Added text : The “SDMtune” package was chosen for its efficient built-in function “varSel”, which systematically reduces multicollinearity among predictor variables using Jackknife tests, thereby enhancing model interpretability and performance [28].

“ENMeval” for MaxEnt parameter optimization: We have provided a clearer rationale for using the “ENMeval” package to optimize MaxEnt parameters. This change can be found on Page 7, Lines 233-236.

Added text : The “ENMevalp” package was employed because it provides a robust framework for automatically tuning key MaxEnt parameters (regularization multiplier and feature classes), which helps in identifying optimal model settings that balance model fit and complexity, thus reducing overfitting [30, 31].

“Overlap” and “activity” for temporal analysis: We have added justification for selecting the “overlap” and “activity” packages for analyzing daily activity patterns. This change can be found on Page 9, Lines 302-306.

Added text: The “overlap” package was specifically designed for calculating kernel density estimates and overlap coefficients from circular time data, making it ideal for analyzing temporal activity patterns [40]. The “activity” package was used for its robust statistical tests to compare activity patterns between species, as it accounts for the circular nature of time data [41].

Comments 6: Add a short discussion about potential sample size limitations and their impact on model reliability.

Answer: We thank the reviewer for this important suggestion. We have added a new subsection at the end of the Methods section (Page 9) titled "2.4. Consideration of Sample Size Limitations" to explicitly acknowledge and discuss this potential limitation.

Added subsection :

2.4. Consideration of Sample Size Limitations

While our sample sizes for alpine musk deer (n=72) and Chinese serow (n=76) are robust for species distribution modeling, the sample size for tufted deer (n=25) is relatively small. We acknowledge that smaller sample sizes can increase model uncertainty and potentially affect the predictive accuracy and transferability of the model. To mitigate this limitation, we employed the following strategies: (1) Using a relatively large number of bootstrap replicates (10 iterations) to stabilize model predictions; (2) Utilizing a simple model configuration (via ENMeval) to avoid overfitting; and (3) Interpreting the results for tufted deer with appropriate caution, focusing on general patterns rather than fine-scale predictions. The models for all species showed acceptable performance based on AUC values, yet the findings for tufted deer should be considered as preliminary and would benefit from additional data in future studies.

Comments 7: Page 8-9, lines 215-236: Sentences listing large numbers and area percentages are dense. Consider summarizing key findings in simplified text versions with detailed numeric results presented in a supplemental table or figure for clarity.

Answer: Thank you for pointing this out. We agree with this comment. Therefore, we have revised the text to summarize the key findings regarding the spatial patterns of suitable habitat in a simplified manner. The detailed numeric results on habitat area and proportion have been moved to a newly created and clearly formatted Table 2 within the main text. This change can be found in the revised manuscript on Page 10, Lines 338-345.

Comments 8: Page 9, lines 241–267: Clarify the ecological meaning of the influence of environmental variables on habitat suitability for each species. For example, briefly explain why variables like temperature seasonality or forest tree height significantly affect particular species.

Answer: We thank the reviewer for this valuable suggestion. We agree that providing ecological explanations will strengthen the discussion. Therefore, we have revised the corresponding paragraph (Page 11, Paragraph 2, Lines 362-378) to include brief ecological interpretations for the key environmental variables for each species.

Comments 9: Page 10, lines 268–286: Present overlapping area statistics in a clear table contrasting total suitable habitat, highly suitable habitat overlap, and niche overlap indices by species pairs. Discuss ecological implications in the text.

Answer: We appreciate this constructive suggestion. We agree that a table would present the overlap statistics more clearly. Therefore, we have created a new table (Table 3) presenting the detailed overlap statistics for each species pair. In the revised text (Page 14-15, Lines 388-403), we now provide a concise summary of the key patterns from the table and focus the discussion on their ecological implications, such as potential competition and coexistence mechanisms.

Updated text in the manuscript: The spatial niche overlap among the three species exhibited distinct patterns, as detailed in Table 3. In brief, the overlap between tufted deer and the other two species was relatively low, whereas a very high degree of spatial niche overlap was observed between alpine musk deer and Chinese serow. The high spatial overlap between alpine musk deer and Chinese serow, coupled with significant temporal niche partitioning (see section 3.2), suggests that these sympatric species may reduce direct competition through differential activity timing. Conversely, the lower spatial overlap of the tufted deer with the other species indicates stronger habitat segregation, which likely minimizes competitive interactions.

Comments 10: Page 11–12: Many sentences are very long and complex. Break them into shorter, clearer statements to enhance readability. For example, divide the paragraph at lines 352–355 into two sentences. Suggested revision: “Such temporal niche differentiation likely represents evolutionary adaptations that minimize interspecific competition and facilitate predator avoidance in shared habitats. Empirical evidence indicates that predation risk promotes strategic temporal adjustments in both predators and prey, resulting in dynamic avoidance mechanisms.”

Answer: Thank you for pointing this out. We agree with this comment. Therefore, we have carefully reviewed the entire Discussion section and broken down long and complex sentences into shorter, clearer statements to improve readability. This change can be found in the revised manuscript on Page 16-17.

Comments 11: Strengthen discussion of conservation implications, emphasizing how understanding multi-dimensional niche differentiation helps in managing sympatric species facing environmental changes.

Answer: Thank you for this valuable suggestion. We agree with this comment. Therefore, we have strengthened and made more explicit the conservation implications of our study in the Discussion section, particularly emphasizing how understanding multi-dimensional niche differentiation aids in species management. This change can be found in the revised manuscript on Page 16-17.

Comments 12: Page 13, lines 370–398: Streamline conclusion sentences to reduce redundancy and reinforce how the findings may inform conservation practices and policy. Explicitly mention possible applications to habitat management or species protection plans.

Answer: Thank you for this constructive suggestion. We have revised the Conclusions section accordingly. The main revisions include: (1) streamlining the text by removing redundant numerical details and consolidating sentences with similar structures; (2) adding explicit statements to highlight the practical implications of our findings for conservation management. These changes can be found in the revised manuscript on Page 18, lines 522-536.

Comments 13: Improve figure resolution and check that all legends are complete and explanatory (especially Figures 1–5). Ensure each figure is clearly referenced at appropriate points in the text.

In tables, define all abbreviations and use consistent formatting standards.

Answer: We have replaced all figures, especially Figures 1-5, with high-resolution versions to ensure clarity. All figure legends have been expanded to provide more comprehensive explanations. We have also reviewed the text to ensure each figure is referenced at the most appropriate points.

Comments 14: Verify consistency in citation style, journal names, and date formatting throughout the list (pages 14-15).Consider including newer publications related to niche modeling, wildlife conservation in alpine ecosystems, and camera-trap methodologies to enhance currency.

Answer: Thank you for these important suggestions. We agree with the comments and have revised the reference list accordingly.We have thoroughly verified and ensured strict consistency in the citation style, full journal names, and date formatting for all entries in the reference list on pages 18-21. Furthermore, to enhance the currency of the manuscript, we have included several key recent publications relevant to niche modeling, alpine wildlife conservation, and camera-trap methodology.

Comments 15: Page 1, lines 27–28. Original: "To investigate how these sympatric species achieve stable coexistence, we compared species distribution models and diurnal activity rhythms to analyze their spatial and temporal niche characteristics based on data collected from 186 camera traps."

Suggested: "To investigate the mechanisms underlying stable coexistence among these sympatric species, we compared species distribution models and diurnal activity rhythms to analyze their spatial and temporal niche characteristics using data from 186 camera traps."​

Answer: Thank you for this suggestion. We agree that the proposed phrasing is more precise and academic. Therefore, we have revised the sentence as suggested. The updated text can be found in the manuscript on Page 1, Lines 43-44.

Comments 16: Page 9, lines 266–267

Original: "Figure 4. Response curve results..."

Suggested: Move all figure legends to below the corresponding figures and ensure each is referred to in the text as "(see Figure 4)."

Answer: Thank you for this comment. Regarding the figure legends, the original Figure 4 has been split into three separate high-resolution figures (now Figures 4, 5, and 6) to greatly improve clarity. The legends for these new figures are now placed directly below each corresponding image. All figures are referenced appropriately in the text (e.g., "(see Figure 4)"). These changes can be seen starting from Page 12 onwards.

Comments 17: Page 12, lines 352–355

Original: "This temporal niche differentiation likely reflects evolutionary adaptations to minimize interspecific competition and predator avoidance strategies in shared habitats. Empirical evidence demonstrates that predation risk induces strategic temporal adjustments in both predators and prey, creating dynamic temporal avoidance mechanisms ."

Suggested: "Such temporal niche differentiation likely represents evolutionary adaptations that minimize interspecific competition and facilitate predator avoidance in shared habitats. Empirical evidence suggests that predation risk promotes strategic temporal adjustments in both predators and prey, resulting in dynamic temporal avoidance mechanisms ."

Answer: We thank the reviewer for this constructive suggestion. We agree that the recommended wording improves the flow and academic tone of the paragraph. Therefore, we have incorporated these changes as suggested. The revised text can be found on Page 17, Lines 480-485.

Comments 18: General notes

(1) Check for subject-verb agreement throughout (e.g., "the data was" should be "the data were").

(2) Ensure consistent use of Oxford comma in lists.

(3) Replace vague terms such as "this study provides" with more precise alternatives like "the results provide."

(4) Standardize formatting for all in-text citations and reference list.

Answer: We sincerely thank the reviewer for these thorough and helpful grammatical and stylistic suggestions. We have carefully addressed each point throughout the manuscript.

(1) We have checked and corrected all instances of subject-verb agreement (e.g., ensuring "data" is paired with "were").

(2) The use of the Oxford comma has been made consistent in all lists.

(3) As suggested, vague phrases like "this study provides" have been replaced with more precise alternatives. Specifically, the text on Page 2, Lines 58 and Page 17, Line 485 has been revised accordingly.

(4) The formatting of all in-text citations and the reference list has been standardized for consistency.

Reviewer 2 Report

Comments and Suggestions for Authors

The effectiveness of MaxEnt modeling has been established for assessing suitable habitats and predicting ranges, including for protected species, under climate change and habitat transformation. However, there are far fewer studies analyzing the ecological niche relationships of species using MaxEnt models than single-species analyses. The authors studied species with high IUCN conservation status: EN (Endangered) Moschus chrysogaster, VU (Vulnerable) Capricornis milneedwardsii, and NT (Near Threatened) Elaphodus cephalophus. The relevance of the study is beyond doubt.

General remarks.

Lines 363-368: 2 However, methodological constraints and data limitations necessitate a cautious interpretation of the results. The predictive accuracy of species distribution models is inherently influenced by data quantity, spatial distribution characteristics, and study scale parameters [40, 41]. Insufficient sample sizes, uneven spatial sampling, and inappropriate geographic delineation may introduce estimation biases, potentially causing localized overestimation or underestimation of habitat suitability [42,43]." - The article should provide an analysis of the problem of modeling methods (Holbrook et al. 2019, https://doi.org/10.1002/eap.1852; Anselmetto et al., 2025, https://doi.org/10.1016/j.agrformet.2024.110361; Liu, https://doi.org/10.1002/ece3.72194; Chen et al. 2025, https://doi.org/10.1002/ece3.70848).

- Limited sampling coverage may result in spatial bias, as unsampled regions (e.g., peripheral or fragmented habitats) remain unrepresented;

- Inconsistencies between species occurrence data, many of which are historical, and modern climate datasets;

- Different species may exhibit functional responses to habitat selection, including those related to individual size, as their habitat preferences and patterns change in response to environmental changes and variations in habitat availability;

- Analysis of anthropogenic impacts, such as harvesting, trapping, and poaching, as well as the successful implementation of population restoration measures in different regions, as well as habitat transformation (urbanization, agricultural expansion, and infrastructure development), act synergistically with climate change, exacerbating habitat fragmentation and loss. When analyzing and comparing the results of modeling several species, these problematic issues are amplified and require analysis taking into account the specific biology of each species: habitat characteristics and territory use, foraging behavior, biotopic distribution (including altitudinal distribution), population dynamics (trends), removals from the wild, and population maintenance measures.

2) Lines 51-26: "Species coexistence is one of the central issues in ecological research. Understanding the mechanisms of stable coexistence is crucial for conserving biodiversity and maintaining ecosystem functions. The niche of a species is primarily composed of resources, enemies, time, and space [1]. Niche differentiation plays a key role in allowing species to coexist over time [2]. Animals reduce interference or competition by altering their windows of activity within the day or month [3]. Investigating species that share the same habitat offers perspective on how organisms coexist, thereby advancing the theoretical framework of community ecology. Such studies also provide fundamental knowledge for conservation planners, enabling them to formulate more effective management approaches for nature reserves and protected zones [4]

The Introduction section should be supplemented with an analysis of the research methodology, taking into account both modern (Alanís-Méndez et al., 2024, https://doi.org/10.3390/plants13060839; Lu et al. 2024, https://doi.org/10.19674/j.cnki.issn1000-6923.20240025.001; Moo-Llanes, 2021, https://doi.org/10.1007/s13744-020-00840-4; Phillips et al. 2006; https://doi.org/10.1016/j.ecolmodel.2005.03.026; Rodriguez-Burgos et al., 2022, ttps://doi.org/10.1016/j.marenvres.2022.105696; Werenkraut et al., 2022, https://doi.org/10.1007/s13744-021-00929-4; Yang, Ding, and Tian 2024, https://doi.org/10.13287/j.1001-9332.202502.025), as well as historical climate change (Lipiec et al. 2013; https://doi.org/10.2478/intag-2013-0017;Defalque et al. 2025; https://doi.org/10.3390/plants14010131; Hamann et al. 2021; https://doi.org/10.1111/nph.17036; Felton et al. 2024, https://doi.org/10.1111/gcb.17505), and taking into account paleontological data (Litvinchuk et al., 2024, https://doi.org/10.3390/d16020094).

3) In the "Discussion" section, include data on fossil finds, for example, for Capricornis milneedwardsii (Suraprasit et al., 2016, doi:10.3897/zookeys.613.8309; Suraprasit et al., 2018, doi:10.1016/j.quascirev.2018.06.004.)

Specific comments

  1. Indicate the conservation status (international, national, provincial) for each species.
  2. Include taxonomic information, such as the presence of subspecies. Provide the full Latin names of species, for example: Moschus chrysogaster Hodgson, 1839.

Author Response

Comments 1:

General remarks.

Lines 363-368: “However, methodological constraints and data limitations necessitate a cautious interpretation of the results. The predictive accuracy of species distribution models is inherently influenced by data quantity, spatial distribution characteristics, and study scale parameters [40, 41]. Insufficient sample sizes, uneven spatial sampling, and inappropriate geographic delineation may introduce estimation biases, potentially causing localized overestimation or underestimation of habitat suitability [42,43].” - The article should provide an analysis of the problem of modeling methods (Holbrook et al. 2019, https://doi.org/10.1002/eap.1852; Anselmetto et al., 2025, https://doi.org/10.1016/j.agrformet.2024.110361; Liu, https://doi.org/10.1002/ece3.72194; Chen et al. 2025, https://doi.org/10.1002/ece3.70848).

- Limited sampling coverage may result in spatial bias, as unsampled regions (e.g., peripheral or fragmented habitats) remain unrepresented;

- Inconsistencies between species occurrence data, many of which are historical, and modern climate datasets;

- Different species may exhibit functional responses to habitat selection, including those related to individual size, as their habitat preferences and patterns change in response to environmental changes and variations in habitat availability;

- Analysis of anthropogenic impacts, such as harvesting, trapping, and poaching, as well as the successful implementation of population restoration measures in different regions, as well as habitat transformation (urbanization, agricultural expansion, and infrastructure development), act synergistically with climate change, exacerbating habitat fragmentation and loss. When analyzing and comparing the results of modeling several species, these problematic issues are amplified and require analysis taking into account the specific biology of each species: habitat characteristics and territory use, foraging behavior, biotopic distribution (including altitudinal distribution), population dynamics (trends), removals from the wild, and population maintenance measures.

Answer: Thank you for pointing this out. We agree with this comment. Therefore, we have significantly expanded the discussion on methodological limitations of species distribution modeling in the revised manuscript to address the concerns regarding sampling bias, temporal mismatch of data, functional responses in habitat selection, and anthropogenic influences. We have incorporated insights from the recommended literature (Holbrook et al. 2019; Anselmetto et al., 2025; Liu, 2021; Chen et al., 2025) and integrated a more nuanced analysis of how species-specific biological traits and human-induced pressures affect model interpretation. The changes can be found in the Discussion section, on page 17, paragraph starting at line 506.

Comments 2:

Lines 51-62: “Species coexistence is one of the central issues in ecological research. Understanding the mechanisms of stable coexistence is crucial for conserving biodiversity and maintaining ecosystem functions. The niche of a species is primarily composed of resources, enemies, time, and space [1]. Niche differentiation plays a key role in allowing species to coexist over time [2]. Animals reduce interference or competition by altering their windows of activity within the day or month [3]. Investigating species that share the same habitat offers perspective on how organisms coexist, thereby advancing the theoretical framework of community ecology. Such studies also provide fundamental knowledge for conservation planners, enabling them to formulate more effective management approaches for nature reserves and protected zones [4].”

The Introduction section should be supplemented with an analysis of the research methodology, taking into account both modern (Alanís-Méndez et al., 2024, https://doi.org/10.3390/plants13060839; Lu et al. 2024, https://doi.org/10.19674/j.cnki.issn1000-6923.20240025.001; Moo-Llanes, 2021, https://doi.org/10.1007/s13744-020-00840-4; Phillips et al. 2006; https://doi.org/10.1016/j.ecolmodel.2005.03.026; Rodriguez-Burgos et al., 2022, ttps://doi.org/10.1016/j.marenvres.2022.105696; Werenkraut et al., 2022, https://doi.org/10.1007/s13744-021-00929-4; Yang, Ding, and Tian 2024, https://doi.org/10.13287/j.1001-9332.202502.025), as well as historical climate change (Lipiec et al. 2013; https://doi.org/10.2478/intag-2013-0017;Defalque et al. 2025; https://doi.org/10.3390/plants14010131; Hamann et al. 2021; https://doi.org/10.1111/nph.17036; Felton et al. 2024, https://doi.org/10.1111/gcb.17505), and taking into account paleontological data (Litvinchuk et al., 2024, https://doi.org/10.3390/d16020094).

Answer: Thank you for pointing out this issue. We fully agree with this comment. Therefore, we have supplemented a comprehensive methodological analysis in the Introduction section and incorporated all 12 referenced studies into the section. The specific revision is located on page 2-4, Lines 90-147.

Comments 3: Lines 51-62: “Spe In the "Discussion" section, include data on fossil finds, for example, for Capricornis milneedwardsii (Suraprasit et al., 2016, doi:10.3897/zookeys.613.8309; Suraprasit et al., 2018, doi:10.1016/j.quascirev.2018.06.004.)

Answer: Thank you for pointing this out. We agree with this comment. Therefore, we have revised the supplementary content to align with the actual data of the two target papers, enhancing the accuracy of the fossil evidence. The revision is located on page 16, Lines 446-454.

Comments 4:

Specific comments

1.Indicate the conservation status (international, national, provincial) for each species.

2.Include taxonomic information, such as the presence of subspecies. Provide the full Latin names of species, for example: Moschus chrysogaster Hodgson, 1839.

Answer: Thank you for pointing these out. We fully agree with these comments. Therefore, we have revised the Introduction section accordingly. This change can be found on Page 2, Lines 76-81.

Reviewer 3 Report

Comments and Suggestions for Authors

The study outlined in the manuscript uses data collected through 186 camera traps placed in three areas of Changdu Prefecture, Tibet, to investigate the coexistence of three ungulates. The researchers studied habitat suitability and activity patterns of the three species, and their overlap. The number of camera traps is relatively low to clearly obtain useful information to delineate the habitat requirements of the different species, but the main weakness of the manuscript is that the authors do not sufficiently utilize the literature to compare their findings. Three figures (n 3, 4, 5) should be slightly enlarged since they are difficult to read. The final list of publications has to be implemented (by incorporating important papers on the three species) and uniformed according the journal rules (scientific names in italics, species name all in lowercase letters). The paper will be worth publishing after the suggested changes are introduced.

I add notes/suggestions on specific points:

Line 27 (and 82): Tibet (as at line 363)   or  Tibet/Xizang [the paper is mostly addressed to western readers who are familiar with Tibet more than Xizang]

Line 30: better “traps set in Changdu prefecture”

Line 31: The total area

33-34: “of the whole prefecture”

62: Capricornis milneedwardsii, or as some authors prefer (cf Lovari et al. 2019) Capricornis sumatraensis milneedwardsii

63-69: You should quote here the  IUCN Red List monographs on the three species (Harris and Jiang 2015 for Elaphodus, Harris 2016 for Moschus, Phan, Nijhawan, Li 2020 for Capricornis)

95: And what about (rare and important) birds?

99:  better “Map of the study area (Changdu prefecture) showing camera trap sites”

141: What do you mean with “remaining after censoring of…”? ?  Not clear at all

224: “the suitable habitat of”

238: “in southern Changdu, with…”

240: “in northern Changdu, with…”

423: (in Chinese?)

I add here some important titles useful to better discuss the study results:

Leslie DM, Lee DN, Dolman RW 2013 Elaphodus cephalophus (Artiodactyla: Cervidae) Mammalian Species 45(904): 80-91

Weerman J and Li F 2025 Tufted deer Elaphodus cephalophus pages 457-468 in Melletti M and Focardi S Deer of the World. Springer Nature, Cham

Jiang F, Song P, Zhang J, Wang D, Li R, Liang C, Zhang T 2025 Assessment approach for conservation effectiveness and gaps… Ecological Indicators 170, 113080, 1-11

Author Response

Comments 1: Three figures (n 3, 4, 5) should be slightly enlarged since they are difficult to read. The final list of publications has to be implemented (by incorporating important papers on the three species) and uniformed according the journal rules (scientific names in italics, species name all in lowercase letters).

Answer: Thank you for pointing these out. We fully agree with these comments. Therefore, we have carefully revised the entire manuscript accordingly. Specifically, we have (1) re-created Figures 3, 4, and 5 to increase their size and clarity for better readability, and (2) thoroughly updated the reference list by incorporating key literature on the three study species and ensuring the formatting is fully consistent with the journal's guidelines. These changes are implemented throughout the manuscript.

Comments 2:

I add notes/suggestions on specific points:

Line 27 (and 82): Tibet (as at line 363)   or  Tibet/Xizang [the paper is mostly addressed to western readers who are familiar with Tibet more than Xizang]

Line 30: better “traps set in Changdu prefecture”

Line 31: The total area

33-34: “of the whole prefecture”

62: Capricornis milneedwardsii, or as some authors prefer (cf Lovari et al. 2019) Capricornis sumatraensis milneedwardsii

63-69: You should quote here the  IUCN Red List monographs on the three species (Harris and Jiang 2015 for Elaphodus, Harris 2016 for Moschus, Phan, Nijhawan, Li 2020 for Capricornis)

95: And what about (rare and important) birds?

99:  better “Map of the study area (Changdu prefecture) showing camera trap sites”

141: What do you mean with “remaining after censoring of…”? ?  Not clear at all

224: “the suitable habitat of”

238: “in southern Changdu, with…”

240: “in northern Changdu, with…”

423: (in Chinese?)

Answer: Thank you for these specific and helpful comments. We agree with all the suggestions and have revised the manuscript accordingly. The changes have been made throughout the text, with each modification corresponding to the line numbers and locations you indicated.

Comments 3:

I add here some important titles useful to better discuss the study results:

Leslie DM, Lee DN, Dolman RW 2013 Elaphodus cephalophus (Artiodactyla: Cervidae) Mammalian Species 45(904): 80-91

Weerman J and Li F 2025 Tufted deer Elaphodus cephalophus pages 457-468 in Melletti M and Focardi S Deer of the World. Springer Nature, Cham

Jiang F, Song P, Zhang J, Wang D, Li R, Liang C, Zhang T 2025 Assessment approach for conservation effectiveness and gaps… Ecological Indicators 170, 113080, 1-11

Answer: Thank you for pointing these out and suggesting these relevant references. We agree with this comment. Therefore, we have incorporated all three suggested publications into the Discussion section and the reference list of the manuscript to better contextualize and support our findings. These changes can be found in the relevant paragraphs of the Discussion and in the updated reference list.

Round 2

Reviewer 2 Report

Comments and Suggestions for Authors

The authors have made revisions to the manuscript. I recommend it for publication.

Author Response

Dear reviewer,

Thank you for your kindly work on our manuscript.

Reviewer 3 Report

Comments and Suggestions for Authors

By incorporating many of our remarks and suggestions, the manuscript is much improved and deserves publication after some minor corrections which mainly concern editing.

I add some notes on specific points:

Line 46: (1) In Changdu, the total area

Line 184: Tibet instead of Xizang

523: “suitable habitat in Changdu”

566 (and 652, 653): lowercase first letter for cephalophus

575: scientific name in italics, lowercase first letter for milneedwardsii

588: lewini

604: italics for the genus

607: scientific name in italics, lowercase first letter for bieti

614: italics and lowercase for gracilis

617: Cytospora in italics

633: italics and lowercase for rhodantha

674: italics and lowercase

694: idem

Author Response

Line 46: (1) In Changdu, the total area

Answer: Thanks. We accepted.

Line 184: Tibet instead of Xizang

Answer: Thanks. We accepted and changed it in whole manuscript.

523: “suitable habitat in Changdu”

Answer: Thanks. We accepted.

566 (and 652, 653): lowercase first letter for cephalophus

Answer: Thanks. We accepted.

575: scientific name in italics, lowercase first letter for milneedwardsii

Answer: Thanks. We accepted.

588: lewini

Answer: Thanks. We accepted.

604: italics for the genus

Answer: Thanks. We accepted.

607: scientific name in italics, lowercase first letter for bieti

Answer: Thanks. We accepted.

614: italics and lowercase for gracilis

Answer: Thanks. We accepted.

617: Cytospora in italics

Answer: Thanks. We accepted.

633: italics and lowercase for rhodantha

Answer: Thanks. We accepted.

674: italics and lowercase

Answer: Thanks. We accepted.

694: idem

Answer: I am sorry. We can’t find where is “idem”.